# Improvement in Epigenetic Aging Clock Induced by BioBran Containing Rice Kefiran in Relation to Various Biomarkers: A Pilot Study

**DOI:** 10.3390/ijms25126332

**Published:** 2024-06-07

**Authors:** Satoshi Kawakami, Ryo Ninomiya, Yusuke Maeda

**Affiliations:** 1Department of Nutrition, Faculty of Health Care, Kiryu University, Midori 379-2392, Japan; 2Research and Development Department, Daiwa Pharmaceutical Co., Ltd., Tokyo 154-0024, Japan; r-ninomiya@daiwa-pharm.com; 3Maeda Clinic, Okayama 701-0205, Japan; maeda@med-pla.com

**Keywords:** aging, lifestyle, epigenetic aging clock, nutrition, BioBran, rice kefiran

## Abstract

Many lifestyle-related diseases such as cancer, dementia, myocardial infarction, and stroke are known to be caused by aging, and the WHO’s ICD-11 (International Classification of Diseases, 11th edition) created the code “aging-related” in 2022. In other words, aging is irreversible but aging-related diseases are reversible, so taking measures to treat them is important for health longevity and preventing other diseases. Therefore, in this study, we used BioBran containing rice kefiran as an approach to improve aging. Rice kefiran has been reported to improve the intestinal microflora, regulate the intestines, and have anti-aging effects. BioBran has also been reported to have antioxidant effects and improve liver function, and human studies have shown that it affects the diversity of the intestinal microbiota. Quantitative measures of aging that correlate with disease risk are now available through the epigenetic clock test, which examines the entire gene sequence and determines biological age based on the methylation level. Horvath’s Clock is the best known of many epigenetic clock tests and was published by Steve Horvath in 2013. In this study, we examine the effect of using Horvath’s Clock to improve aging and report on the results, which show a certain effect.

## 1. Introduction

Many lifestyle-related diseases such as cancer, dementia, myocardial infarction, and stroke are known to be caused by aging [1]. From 2022, the WHO disease classification system, the ICD-11 (International Classification of Diseases, 11th edition), has included a code for “aging-related” diseases [2]. In other words, although aging is irreversible, aging-related diseases are reversible, and treatments and countermeasures should be taken in order to lead a long and healthy life and prevent other diseases [3].

In recent years, it has become possible to quantitatively measure aging, which correlates with disease risk, through the “epigenetic clock test”, which examines the sequence of all genes and determines biological age from their methylation levels [4]. The most well-known epigenetic clock test is “Horvath’s Clock”, which was announced by Steve Horvath in 2013 as an improved fourth-generation test. DunedinPACE is currently the most reliable test, and it is an excellent test that can also measure the acceleration of aging [4].

Furthermore, it is known that aging progresses not only due to the effects of aging but also due to damage to tissues and cells resulting from inflammation [5]. Interleukin [IL]-1, IL-6, IL-8, tumor necrosis factor [TNF]α, etc., are implicated in one important issue [6].

It is thought that senescent cells become difficult to remove due to the decline in immune system function caused by the aging phenomenon, and thus they are present at increased concentrations in the body [7]. Therefore, aging can be viewed as a problem of immune function [8], as various immune cells act against and try to remove senescent cells, but their effects may be weakened due to aging (Figure 1) [9]. There is a limit to the number of times cells can divide in the human body, and this is called the Hayflick limit [10]. Cells that have reached the Hayflick limit are treated as senescent cells, and the state in which these senescent cells accumulate in the body is called senescence. Under normal circumstances, biological aging would progress due to a decline in immune function, leading to the development of lifestyle-related diseases, etc. [9].

Therefore, an approach to tackling aging is urgently needed, and this time, we investigated improvements to aging. BioBran containing rice kefiran was used in this study. BioBran containing rice kefiran was selected as it activates the body’s NK cells, T cells, B cells, and macrophages through [1] direct blood circulation and [2] stimulation of Peyer’s patches in the ileum, and it also regulates immunity. It is thought to have activating, anti-inflammatory, anti-allergic, anti-oxidant properties, etc. [11,12,13,14], and is highly likely to be useful in improving aging.

Additionally, aging is accelerated by the deterioration of the intestinal environment [15], and BioBran containing rice kefiran is also known to improve the intestinal environment [16]. Additionally, active oxygen is an important factor in aging [17]. Active oxygen originally exists in the granules of neutrophils and is necessary for destroying antigens in living organisms [18]; equally, excessive active oxygen also affects normal cells [19]. This may lead to chronic inflammation [20], which can lead to the progression of aging. In order to live, humans ingest oxygen and use it within mitochondria, which contribute to the production of ATP [21]. In other words, it is possible to produce energy because of the presence of oxygen [22]. However, not all of this oxygen is used and only a small percentage becomes active oxygen [23]. Active oxygen oxidizes various sites in the body, causing damage to blood vessel walls and DNA [24], which is believed to lead to the development of vascular diseases and malignant neoplasms [25]. Since active oxygen is closely related to aging, we also considered that BioBran has antioxidant effects [26].

The above results suggest that BioBran containing rice kefiran may be highly effective in preventing aging by enhancing immune system function, preventing the increase in senescent cells, and suppressing inflammatory cytokines.

Here, using the epigenetic clock test from a multifaceted perspective on aging, we aimed to elucidate the rejuvenating effect of ingesting BioBran containing rice kefiran on biological age (epigenetic aging clock) in healthy subjects. It is believed that this research will contribute to the creation of new preventive medical treatment strategies by suppressing aging progression through immune system regulation. In addition, since this is a pilot study, various biomarkers were measured in a small number of subjects. As discussed in the Methods section, we commissioned TruDiagnostic™ (Lexington, KY 40503, USA) and followed the company’s standard methods for measuring various biomarkers [27].

## 2. Results (Table 1, Figure 2)

The mean value, standard deviation, and *p* value of each biomarker are listed in Table 1. Each result is described below.

**Table 1 ijms-25-06332-t001:** Biomarkers average and SD, *p* value. (A) The average unit is “years“, (B) The average unit is “none”.

(A)			
Test Substance	Before	3 Months Later	*p* Value
	Mean ± SD	Mean ± SD	
Chronological age	60.36 ± 6.56	60.49 ± 6.57	
Biological age	65.3 ± 5.52	62.07 ± 6.49	0.036
Telomere age	63.65 ± 8.03	57.99 ± 8.78	0.025
**(B)**			
DunedinPACE	1.04 ± 0.07	0.97 ± 0.13	0.141
Telomere Length	6.89 ± 0.13	6.99 ± 0.14	0.030
DNAm CRP	93.52 ± 6.29	89.15 ± 13.95	0.161
DNAm IL-6	76.21 ± 22.12	57.11 ± 16.84	0.025

**Figure 2 ijms-25-06332-f002:**
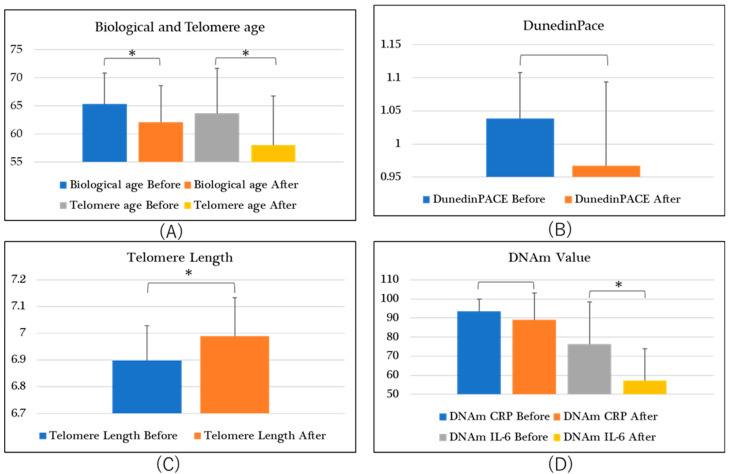
Biomarkers and SD. (**A**) unit is “years”, (**B**–**D**) units are “none”. *, *p* < 0.05.

### 2.1. Biological Age

A Wilcoxon *t*-test was performed for biological age and a significant difference before and after intake was observed at *p* < 0.05 (Table 1A, Figure 2A: Biological Age, Before and After). Therefore, it appears that a decrease in biological age was observed.

### 2.2. DunedinPACE Average

DunedinPACE is the speed of aging progression. When Wilcoxon’s *t*-test was performed on these results, no significant difference was found, but the values tended to be low (Table 1B, Figure 2B: DunedinPACE, Before and After). It is thought that a significant difference in DunedinPACE values will be observed by increasing the number of cases in the future.

### 2.3. Telomere Length

When the test substance was administered and telomere length was measured, a significant difference was observed at *p* < 0.05 (Table 1B, Figure 2C: Telomere Length, Before and After). Therefore, it is possible that an extension of cell division lifespan was observed.

### 2.4. Telomere Age

When the test substance was administered and telomere age was measured, Wilcoxon’s *t*-test showed significance at *p* < 0.05 (Table 1A, Figure 2A: Telomere Age, Before and After), suggesting that a recovery of the telomere itself occurred.

### 2.5. DNAm CRP

DNAm CRP is an indicator of inflammatory responses in DNA. When the test substance was administered, no significant difference was observed (Table 1B, Figure 2D: DNAm CRP, Before and After). However, as a decreasing trend was observed, it is possible that increasing the n number in the future may highlight any significant differences.

### 2.6. DNAm IL-6

DNAm IL-6 is an indicator of IL-6-induced DNA methylation. It is thought that as DNA methylation progresses, the amount of IL-6 secreted decreases, leading to a lack of immune system regulation. When a Wilcoxon *t*-test was performed on the test substance and DNAm IL-6 values, a significant difference was observed at *p* < 0.05 (Table 1B, Figure 2D: DNAm IL-6, Before and After). Therefore, it is possible that the test substance modulates the immune system.

## 3. Discussion

BioBran-containing rice kefiran describes a mixture of rice kefiran and biobran.

Rice kefiran is a fermented product obtained by culturing the lactic acid bacteria Lactobacillus kefiranofaciens, which is isolated from kefir, a traditional fermented milk food from the South Caucasus region, using the enzymatic decomposition products of rice as a nutrient source [28]. It contains a water-soluble polysaccharide called kefiran, which reportedly improves the intestinal flora [16], regulates the intestinal tract [29], and has anti-aging effects [30]. Since aging is said to be accelerated by the deterioration of the intestinal flora, it is thought that improving the intestinal flora may help prevent aging [31]. Furthermore, BioBran is manufactured using water-soluble dietary fiber (hemicellulose B) and comprises approximately 5% of the main raw material of rice bran, which is the hard outer shell discarded during rice milling. Rice bran hemicellulose B is a dietary fiber with a complex structure consisting of arabinose and xylose as constituent sugars, and it is characterized by a relatively small molecular weight. It is not known to have immunostimulatory effects as is. However, there is a report that immunostimulatory effects can be observed when hemicellulose B is partially hydrolyzed with a carbohydrate-degrading enzyme complex obtained by culturing shiitake mycelia [32]. Furthermore, BioBran has been reported to have antioxidant effects [33] and improve liver function [34], and human studies show that it affects the diversity of the intestinal flora [35].

This study suggests that biological age may be improved by ingesting BioBran containing rice kefiran. To begin with, biological age is the in vivo age relative to chronological age and can be used to approach aging [36]. Up until now, biological aging and chronological aging were viewed as being on the same page, but the introduction of the ICD-11 has led to definitions of aging as simply the passage of time or as a disease [2]. In other words, preventing aging can be interpreted as the prevention of aging-related diseases [37]. As mentioned above, aging is simply a passage of time and is irreversible. However, aging can be said to be reversible [38], so this study’s rejuvenation biological age has great significance.

We will discuss the various biomarkers used in this study.

### 3.1. Biological Age (Table 1A, Figure 2A: Biological Age, Before and After)

Biological age represents the internal age of a person in relation to their chronological age and is also used as an indicator of aging [4]; thus, as an indicator of aging, it can describe the reversible process. A significant decreasing trend was observed when the test substance was used. In other words, we suggest that aging can be approached. Aging is the cause of various diseases [39]. Cancers, including malignant neoplasms and vascular diseases associated with aging-related vascular damage, are also thought to be related to aging [40,41]. Therefore, the improvement in biological age observed in this study suggests that it may be possible to prevent diseases associated with aging. The discrepancy between chronological age and biological age has an important meaning, and a younger biological age may be useful not only for preventing diseases but also for maintaining and improving health.

### 3.2. DunedinPACE (Table 1B, Figure 2B: DunedinPACE, Before and After)

DunedinPACE is a measure of the speed of aging [42]. It was originally derived from the Dunedin study, which quantified the rate of aging over a 20-year period from age 26 to 45 in the city of Dunedin on New Zealand’s South Island. It is a numerical representation of whether the progression of cancer can be suppressed or not. In other words, there is a correlation between DunedinPACE and biological age [43]. In this study, although no significant difference was found in DunedinPACE, a decreasing trend was observed. However, since biological age was improved with a significant difference, it can be concluded to have a correlation with the test substance. It is said that aging is suppressed when the DunedinPACE is slowed [44]; therefore, this study suggests that the speed of aging may have been be suppressed, although the difference observed was only a trend.

### 3.3. Telomere Length (Table 1B, Figure 2C: Telomere Length, Before and After)

Telomeres exist at the ends of chromosomes and consist of the base sequence TTAGGG [45]. Somatic cells divide according to the cell cycle, and their telomeres become shorter as they repeatedly divide [46]. Telomeres are also said to prevent damage to DNA and errors during cell division [47].

Eventually, when telomeres cease to exist, the cell itself becomes unable to divide, resulting in the existence of senescent cells [48]. However, telomerase is an enzyme that protects telomeres, and activation of this enzyme is said to lead to telomere protection and elongation [49]. In this study, a significant elongation of the telomeres in blood nucleated cells was confirmed.

Although hypotheses, we believe that the test substance caused the following effects.

① The test substance had an effect on promoting the activation of telomerase.

② The test substance itself has an effect similar to that of telomerase.

③ As a result of the immune system activation, senescent cells were eliminated and the average telomere length was lengthened.

As this is a possibility, we would like to investigate the molecular and physiological behavior of the test substance in the future.

### 3.4. Telomere Age (Table 1A, Figure 2A: Telomere Age, Before and After)

Because telomere age is correlated with biological age, telomere age is calculated as a predicted value from telomere length. Telomeres are generally said to be shorter in older people than in younger people [50]. Therefore, young telomere age is thought to lead to telomere lengthening or protection, as mentioned above [49]. Although no significant difference in telomere age was observed in this study, there was a trend toward improvement in telomere age. Although this is also a hypothesis, it is possible that the test substance had the same effect as above. Therefore, it will be a subject of future investigation.

### 3.5. DNAm CRP (Table 1B, Figure 2D: DNAm CRP, Before and After)

DNAmCRP is an epigenetic measure of C-reactive protein (CRP), which is a signature of DNA methylation and a specific measure of inflammation [51]. It is possible that a decrease in DNAm CRP could reduce inflammatory response in DNA [52]. The DNA inflammatory reaction referred to here is mainly caused by DNA damage resulting from reactive oxygen species [53]. Since aging is said to be caused by a chronic inflammatory response in senescent cells, a decrease in DNAm CRP levels is considered to be an important indicator of aging prevention [54]. Although no significant difference was observed in this study, a decreasing trend was observed, suggesting that DNA methylation may be suppressed by increasing the administration period and n number in the future.

### 3.6. DNAm IL-6 (Table 1B, Figure 2D: DNAm IL-6, Before and After)

DNAm IL-6 is an indicator of IL-6-induced DNA methylation [55]. It is thought that, as DNA methylation progresses, the amount of IL-6 secreted decreases, leading to a lack of immune system regulation [56]. In this study, a significant decrease in DNAm IL-6 was observed, suggesting that the test substance may modulate the immune system. Among the test substances, BioBran was originally used as a substance that regulates the immune system [57]. Until now, it was understood that BioBran regulated the immune system by stimulating Peyer’s patches [58], but this study provides a different perspective, wherein BioBran also reduces DNAm IL-6. This suggests that the immune system may be regulated by enhancing the effect of IL-6 [59].

From the above, it is suggested that in vivo rejuvenation may be realized by using BioBran containing rice kefiran. According to the literature, biological age can be improved by following instructions on diet, exercise, and sleep [60,61,62,63]. However, the interventions in this study are of great significance as the improvement effect observed by using the test substance alone, without any other tests, reduced the burden forced on the subjects.

Until now, aging was considered to be irreversible, but this research has great significance in terms of preventing the diseases associated with aging. Although it is possible to correct the appearance of aging to some extent through cosmetic surgery, it is nearly impossible to correct the appearance of biological aging through cosmetic surgery. Furthermore, although is it widely known that changing one’s eating habits is necessary, it is difficult to make sudden changes to one’s eating habits due to accumulated experiences. It has also been pointed out that sudden changes in diet can lead to stress, causing the secretion of stress hormones such as adrenaline and cortisol, leading to stress-related diseases [62]. Sleep disorders are generally a problem caused by stress [64]. Controlling one’s mental health is extremely important, as a decline in sleep quality can increase risks of anxiety and depression, leading to a decline in QOL [65]. There was no direct causal relationship with stress in this study, but there are various reports in the literature indicating that biological age improved due to sleep-related interventions [66], so no sleep-related intervention was explored in this study. Our result indicates an ability to rejuvenate biological age without stress and is considered to be very clinically meaningful. A search of the literature revealed no data indicating that biological age can be improved by taking the test substance and changing lifestyle habits without stress. This study is the first to show that biological age can be improved by simply consuming the test substance without enforcing lifestyle changes or intervening through exercise or dietary restrictions. It is believed that further elucidation of the mechanism will contribute to extending not only average life expectancy but also healthy life expectancy.

Furthermore, the lifestyle-related diseases that currently pose problems are generally a phenomenon associated with aging and are not simply caused by age. Lifestyle-related diseases account for about half of the causes of death, so preventing the aging phenomenon is considered to be a very important factor for the above reasons. Taking diabetes as an example, blood sugar levels can be raised from various angles such as glucagon, cortisol, and fat burning, but insulin is the only way to lower blood sugar levels. As there can only be one type of insulin in the original human form, blood sugar levels are extremely important. However, people are said to be in an era of satiation these days and can freely eat whatever they want. Furthermore, diabetes can be caused by things like drinking and smoking. On a different note, cancer cells proliferate due to errors in cell division, and when they deposit in organs cancer develops. Active oxygen is said to be deeply involved in this process [67]. Active oxygen can be broadly defined as hydrogen peroxide, superoxide, hydroxyl radicals, and singlet oxygen, which are said to have a deep relationship with aging. Furthermore, since these active oxygen species can increase the risk of cancer, it is also suggested that improving biological age may remove active oxygen species themselves. Diabetes was given as an example as the test substance used in this study also removes active oxygen, as mentioned above [26]. As humans evolve and lifestyles change, it is thought that this test substance may possibly improve the internal environment and cause a return to biological age. By optimizing homeostasis, which keeps the internal environment of the body constant, and improving biological age, there is a possibility of returning the body to its original, correct human form. From this perspective, the significance of this research is considered to be extremely great.

## 4. Materials and Methods

### 4.1. Type of Research

Single-blind functional evaluation.

### 4.2. Approach to Bias

This study population was selected because Japanese men in their 50s and above are at an age when lifestyle-related diseases become more common. It is also an age at which hormonal changes occur in certain women due to menopause, etc. Young people were excluded because they show no significant difference in biological age, and there was no possibility that changes in their internal environment would be significant.

### 4.3. Intervention Method

Ethics approval for this study was obtained through the Ethics Committee of the International Society of Geriatrics and Geriatrics (ISGN_NI10012023).

Of the subjects who applied for this study, we selected 9 healthy people with no underlying diseases and obtained their consent. We administered BioBran containing rice kefiran to the 9 subjects in a single-blind manner. Subjects were orally administered two packets of BioBran containing Rice Kefiran (1000 mg BioBran, 500 mg Rice Kefiran per packet) with water or lukewarm water 30 min before meals or on an empty stomach for 12 weeks. The effectiveness of BioBran containing rice kefiran was verified by measuring the following test items and observation items centered on biological age (described in the results) before and after the start of the test. One subject was negligent in responding to the intervention instructions during the study, so the final evaluation was conducted on eight subjects. Regarding the various biomarkers, we commissioned TruDiagnostic™ (Lexington, KY 40503, USA) to statistically process the test results.

### 4.4. Evaluation Items

(1)Biological age;(2)DunedinPACE;(3)Telomere length;(4)Telomere age;(5)DNAm CRP;(6)DNAm IL-6.

We mainly investigated the above six items. Their biological significance is discussed in the Results and Discussion sections.

The subjects’ information is described below (Table 2).

### 4.5. Evaluation Method

Based on the numerical results taken before and after the intervention, statistical significance was tested using the statistical processing software IBM SPSS Statistics (Ver. 25). In addition, Wilcoxon’s *t*-test was used and the significance level was set at *p* < 0.05.

## 5. Conclusions

In this study, we used BioBran containing rice kefiran to examine whether biological age can be improved. A discrepancy between chronological age and biological age was observed with the use of the test substance, and the results show that biological age improved. As aging is currently treated as a disease according to the ICD-11, the results obtained with this test substance are considered to be useful for the prevention and treatment of aging. In addition, although there are many interventions described in the literature such as diet, exercise, and sleep, a search of the literature found no examples of improvement in biological age without these interventions. In other words, it is suggested that regular consumption of the test substance may lead to the prevention and treatment of aging. Aging is known to cause various lifestyle-related diseases, such as cancer and heart disease. Stress is another cause of aging. It is undeniable that interfering with lifestyle habits can lead to stress and accelerated aging when it would normally be a treatment for aging. One possible cause of this is active oxygen, which is also mentioned in the text. Since the test substance also has the effect of removing active oxygen, removing active oxygen is considered to be possible. In a broad sense, active oxygen includes hydrogen peroxide, superoxide, hydroxyl radicals, and singlet oxygen, which are also known to be deeply involved in aging. Therefore, using the test substance eradicates the need for any special lifestyle interventions, and simply drinking the substance means that there is no stress or invasiveness; in this way, it is more feasible to improve biological age. However, this study has some weaknesses in terms of the study design, such as the small number of subjects and the single-blind protocol. In the future, we would like to elucidate the mechanisms underlying the decline in biological age and the elongation of average telomere length from a molecular and physiological perspective.

## Figures and Tables

**Figure 1 ijms-25-06332-f001:**
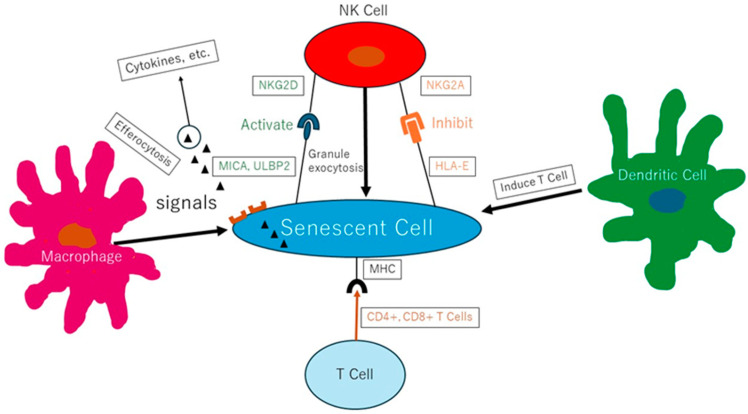
Senescent cell and immune system [9].

**Table 2 ijms-25-06332-t002:** Chronological age average.

No Item	Before	3 Months Later
	Mean ± SD	Mean ± SD
Chronological Age	60.36 ± 6.56	60.49 ± 6.57

## Data Availability

In conducting this research, research funds and test products were provided by Daiwa Pharmaceutical Co., Ltd. (Tokyo, Japan), and collaborative research was conducted. We also received assistance from Daiwa Pharmaceutical Co., Ltd. when preparing this paper.

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
