# Peer review of "Improvement in Epigenetic Aging Clock Induced by BioBran Containing Rice Kefiran in Relation to Various Biomarkers: A Pilot Study"

_ijms, 2024, doi:10.3390/ijms25126332_

Round 1

Reviewer 1 Report

Comments and Suggestions for Authors

Comments to the authors:

In this study, the authors explored the improvement of the epigenetic aging clock by biobran containing rice kefiran. The authors employed BioBran, which contains Rice Kefiran, as a strategy to mitigate aging. Rice Kefiran is noted for enhancing the intestinal microflora, regulating bowel function, and exhibiting anti-aging properties. Similarly, BioBran is recognized for its antioxidant capabilities and its ability to improve liver function. Furthermore, the human studies have demonstrated its impact on the diversity of the intestinal microbiota. Current methods for measuring aging, which correlate with disease risk, include epigenetic clock tests that analyze the entire gene sequence to determine biological age based on methylation levels. The most renowned among these tests is Horvath's Clock, developed by Steve Horvath in 2013. The research focused on assessing the effects of Horvath's Clock on aging. The findings indicate that the intervention had a discernible impact on the aging process. Overall, although this study is some interesting; however, the data provided by the authors can not support their conclusion at the current stage. I have some concerns that need to be addressed by the authors.

1: The study utilizes a single-blind method with a relatively small sample size (only nine participants initially, with final analysis on eight). This limited sample size and single-blind design may introduce bias and limit the generalizability of the findings​​. Future studies would benefit from a double-blind, placebo-controlled design and a larger sample size to enhance validity and reproducibility.

2: The statistical significance in some measures, such as biological age and telomere length, suggests potential benefits of the treatment​​. However, other measures like DunedinPACE showed no significant difference but a trend towards improvement, highlighting the need for a larger study population to potentially reveal significant effects​​.

3: While the study reports significant effects on aging biomarkers following the intake of BioBran containing Rice Kefiran, it does not appear to thoroughly account for or control other lifestyle factors that could influence aging, such as diet, physical activity, and stress levels, although these are acknowledged in the discussion​​. Including control groups that account for these variables could strengthen future research outcomes.

4: The findings are derived from a specific demographic (Japanese men aged 50 and above and women experiencing menopausal hormonal changes), which may not be applicable to other populations or age groups​​. Expanding the demographic scope in future studies could enhance the relevance and applicability of the results.

5: The study was funded and supported by Daiwa Pharmaceutical Co., Ltd., which also provided the test substance. Although the study claims no direct role of the company in the study design or result analysis, this relationship could lead to potential conflicts of interest that might influence study outcomes​​.

6: The paper discusses potential mechanisms by which BioBran containing Rice Kefiran could influence aging markers, but these are largely speculative. More detailed molecular or physiological studies would be beneficial to clearly define how these changes occur and to validate the biological plausibility of the observed effects​​.

7: The use of specific biomarkers (e.g., DNAm CRP, DNAm IL-6) provides a focused insight into the molecular changes potentially induced by the intervention. However, the lack of significant changes in some of these markers suggests that the effects might be more limited or variable than the study suggests​​.

In summary, while the study presents intriguing preliminary data on the potential anti-aging effects of BioBran containing Rice Kefiran, there are several areas where methodological improvements could enhance the robustness and reliability of the findings.

Comments on the Quality of English Language

 Moderate editing of English language required

Author Response

1.ご指摘の通り、今回はプラセボ群がなかったので、今後対応していきたいと考えています。そこで今回は、記事ではなく、事例報告というパイロットスタディに変更したいと思います。
2.また、症例数が少ないため制限があるという文言を追加しました。
3.英文はネイティブスピーカーがチェックしていますので、ご確認ください。

Reviewer 2 Report

Comments and Suggestions for Authors

In this study the authors suggest that biobran-containing kefiran rice improves biological age, as estimated using several parameters (e.g. epigenetic clock, telomere length, etc).

This seems an interesting idea, but I am afraid not well presented in the current form of the manuscript to justify publication.

1. The manuscript needs to be re-checked for grammar and syntax errors and phrasing (for examples, please refer to the uploaded file).

2. The term senescence abruptly appears in the text with no introduction. The authors should devote a part referring to the characteristics of senescent cells and to the association of their accumulation in the tissues with the manifestation of aging-related diseases.

3. In Table 1 (p13) it is not clear to me how chronological age can change by any treatment.

4. The authors should re-write the Materials and Methods section describing in detail how all parameters appearing to be measured were actually experimentally calculated. I assume that the authors have used blood cells to extract DNA. How DNA methylation was assessed? How telomere length was measured? And so on. Please elaborate.

5. I feel that the number of donors (9) is very small to draw solid conclusions. The study should be extended using a higher number of cases.

Comments on the Quality of English Language

The manuscript needs to be re-checked for grammar and syntax errors and phrasing (for examples, please refer to the uploaded file).

Author Response

1.Regarding grammar errors, the English text has been checked by a native speaker, so please check it.

2.Please note that we have corrected the issue where the word senescent suddenly appeared.

3. I have added a note regarding chronological age.

4.The method for measuring methylation, etc. was outsourced, so I couldn't learn more about it in depth, but I found a paper that uses this company to measure methylation, etc., so I'd like to add it. I did.

5.Since the number of cases is certainly small this time, I have changed the title to "Case Report Pilot Study."

Thank you for your understanding.

Round 2

Reviewer 1 Report

Comments and Suggestions for Authors

The authors have addressed all the concerns

Author Response

Thank you for choosing yes.
Another reviewer corrected the wording in one place, so I made the corrections.
Thank you for this time.

Reviewer 2 Report

Comments and Suggestions for Authors

This is the revised version of a previously submitted manuscript. The authors have attempted to answer to some of my queries raised during the previous round of the reviewing process, but I am afraid the study still does not meet the standards required for its publication in a high-impact Journal such as IJMS.

Given the lack of functional experiments and the low number of included subjects, I feel that the authors should attempt to transfer their pilot study to another Journal.

Specific point: Ln50-52: Please correct helix limit to Hayflick limit.

Comments on the Quality of English Language

English has been improved in the revised version of the manuscript, but there are overseen mistakes, e.g. the renowned term in the filed of senescence "Hayflick limit" appears as helix limit.

Author Response

Thank you for pointing this out.

At this stage, this is a pilot study, but we would like to conduct an RCT test as a second report and submit it to this journal.
Therefore, I would like to publish this article in this journal.
I also took a native language check, and I have made corrections to the words that were pointed out to me. Thank you for your understanding.
